# Social Network Analysis as a Tool in the Care and Wellbeing of Zoo Animals: A Case Study of a Family Group of Black Lemurs (*Eulemur macaco*)

**DOI:** 10.3390/ani13223501

**Published:** 2023-11-13

**Authors:** Max Norman, Cassie Jones, Kara Watson, Renato L. Previdelli

**Affiliations:** 1Department of Animal Management and Saddlery, Capel Manor College, London EN1 4RQ, UK; 2Paradise Wildlife Park, Zoological Society of Hertfordshire, Broxbourne EN10 7QA, UK; cassie.j.jones@outlook.com (C.J.); research@pwpark.com (K.W.); 3Comparative Biomedical Sciences, Royal Veterinary College, London NW1 0TU, UK; rprevidelli@rvc.ac.uk

**Keywords:** social network analysis, animal management, animal welfare, zoos and aquariums, primates, black lemur, social behaviour

## Abstract

**Simple Summary:**

Social network analysis (SNA) has the potential to be a vital element of a holistic animal care toolbox, which allows in-depth examination of the social roles of individuals and the dynamics at a group level. By studying a case study of a group of four (1.1.2) black lemurs (*Eulemur macaco*) living in human care at a zoo in the United Kingdom, the potential for SNA as a practical tool in assessing animal wellbeing is presented. SNA of the lemur group revealed patterns of female dominance, with one individual male being the most frequent target of aggressive interactions from the group. Using this technique, animal caregivers can identify social pressures, monitor social dynamics and dominance, and examine causes of suboptimal social wellbeing among the individuals living in their care. Understanding the social networks of individual groups can facilitate improvements and interventions to the benefit of individuals, groups, and wider populations.

**Abstract:**

Social network analysis (SNA) is an increasingly utilised technique in the literature examining the social structures and organisation of animals and understanding the bonds between groups and individuals. Using a case study as an illustration, the applications of SNA are explored, including the identification of dominance hierarchies and detection of sources of social pressure, with a particular focus on the applications of SNA to holistic assessments of animal welfare alongside other methods. Based on the examination of social dynamics in a family group of four black lemurs (*Eulemur macaco*), a primate whose social organisation is characterised by patterns of female dominance, it is demonstrated that SNA can be used to examine the affiliative and agonistic interactions between individuals living in human care. SNA showed species-typical forms of female dominance that were largely directed towards the two males, characterised by the initiation of aggressive interactions and male submission. More intricate relationships and consistent social roles across networks were revealed through the examination of SNA. It is concluded that SNA has wide-ranging benefits in the assessment of effects of environmental changes, such as informing social management decisions, developing enrichment and intervention programs, and guiding overall improvements to the housing and care of individual animals. SNA, as part of an animal welfare toolbox, could, therefore, be a pivotal technique for modern animal welfare assessment that considers individual animals and their social lives. By sharing a case study of the technique in use, it is hoped that animal collections may adopt similar modern and evidence-based assessment methods.

## 1. Introduction

Social relationships are essential to the wellbeing of many species living in zoos and aquariums worldwide. The bonds that animals form with each other provide numerous benefits to their physical and psychological health, and as such, many modern models of animal welfare consider the complex implications that the social lives of animals can have on individual wellbeing. The Five Domains model is a well-known modern framework for assessing animal wellbeing, highlighting the five key areas of nutrition, environment, health, behaviour, and mental states as essential to good animal welfare [1]. This model emphasises the importance of examining the interface of all areas of an animal’s life, including their interactions with other animals, in terms of influencing mental states [1]. Similar frameworks for animal welfare assessment highlight a similar connection, such as the expanded Welfare Quality criteria for wild animals in human care and the 24/7 across-lifespan framework, which underlines the responsibility of those caring for animals to provide opportunities for natural, non-harmful social behaviours and positive emotional states [2,3].

Good welfare encompasses all interconnected elements of an animal’s physical and psychological state, including their perceptions of the social environment. For brevity, the terms welfare and wellbeing are used synonymously within this paper. Maintaining and promoting appropriate social groupings and interactions that are positive, beneficial, and meaningful from the perspective of the animal is, therefore, a key challenge for zoos and aquariums, where the human-managed environment can impact the social behaviours of animals in unexpected and complicated ways. As, when combined with all other areas of their lives, the social environment is important to physical and psychological health [4], it stands to reason that attention should be paid to measuring the social experience of animals in conjunction with all other areas of their lives when assessing wellbeing. The development and maintenance of social bonds are essential aspects of behaviour for many species, and stable social groups are known to be positive and supportive environments [5,6]. Positive social interactions, such as play, social grooming, and socially resting in contact, promote positive wellbeing [3], with numerous other benefits, such as cooperation towards common goals and the promotion of physical and psychological wellbeing, also noted [7,8,9]. However, there are risks associated with group living that can be challenging to manage in human-managed contexts, such as zoos, where there are fewer options for animals to make social choices for themselves [10]. For example, aggression may be driven by access to resources such as food, and, indeed, aggression is often higher during feeding periods in both free-living animals and animals living in human care [11,12,13]. 

While some level of aggression is unavoidable, as it is a necessary component of the maintenance of group cohesion and dominance structure [14,15,16,17], individuals who are frequently targets of anti-social behaviours often experience poorer wellbeing that is exacerbated by the zoo environment, where there are fewer opportunities to escape from aggressors [18,19]. Aggressive or more dominant animals may restrict subordinates’ access to food, shelter, and other resources [20,21], and animals who are isolated from their social group are unlikely to receive the positive benefits of sociality, instead experiencing stress and loneliness [22,23]. Of course, an understanding of these negative effects on individuals must also be balanced with the wellbeing of the group as a whole and recognition that attempting to intervene to the benefit of those individuals who are subordinate can have wider detrimental impacts on group stability [24]. An ability to understand these complex interactions within social groups is integral to predicting the impacts of different management decisions on the care of group-housed animals and optimising individual and group-level wellbeing.

Traditional approaches to measuring social behavioural patterns for management purposes include assessing proximity to others [25,26] and observational assessments of rates of agonistic and affiliative interactions at the individual or group level [26]. Social network analysis (SNA) has become increasingly popular as a more complex and holistic tool for quantifying and visualising the complexity of social environments, both in the wild and when living in human care, based on behavioural observations [27]. SNA is an effective method of quantifying patterns of sociality [10,28,29] and offers a complementary approach to traditional measurements of social behaviour, providing a more in-depth and holistic assessment of the relationships and individual roles present in social groups [10,30,31]. By mapping which individuals are interacting with others, who is initiating different types of social behaviours, and with what frequency these behaviours are occurring, SNA can reveal social roles, affiliations, and possible sources of tension within a group, as well as which connections are most important to individuals and the stability of the whole group [10]. 

SNA has been presented as a valuable tool for the management of zoo animals [10] owing to the simplicity of collecting relevant data [32]. In allowing researchers to understand the intricacies of social networks within groups of animals, it may become possible to predict and better manage the social environment. For example, translocating animals out of social groups for conservation breeding programmes is a common practice in zoos and aquariums, though the impacts of disrupting current social groupings can be unpredictable and complex. Using SNA, it may be possible to determine which individuals are most important or central to the social environment and, thus, which animals are more or less likely to significantly impact the dynamics of the group if removed. Mapping changes to social networks over time may also be broadly useful in tracking, for example, the changing social roles of animals as they progress through their lives. Understanding how age impacts social role would be particularly useful to conservation breeding programs such as European Ex-situ Programs (EEPs) and Species Survival Plans (SSPs), where the most appropriate ages for dispersal from natal social groups may not be known for some species.

With appropriate training, SNA can be conducted by student researchers, visiting academics, and animal caregivers themselves, with simple and practical graphical user interfaces for conducting the analyses, such as UCINET and NetDraw, being available. For animals living in zoos and aquariums, evidence-based insights into social structures can be essential in terms of guiding management strategies to improve animal wellbeing, management outcomes, and other goals, such as conservation breeding success [10,33]. A recent review of zoo-based SNA research has, however, highlighted that while SNA has been presented as a valuable tool in many studies, many published reports are basic and largely descriptive, with less focus on applied case studies [32]. Many of the case studies that have been published with regard to animal care actions focus on the impacts of these actions on social networks. Existing case studies examine how social networks shift and change as a result of changes in group composition [33,34,35,36,37] and the impacts of moving animals to a new habitat [38]. While these studies are indeed essential for understanding how social groups are impacted by care activities, less focus has been given to demonstrating how SNA can be used in a predictive or preventative capacity as part of a holistic animal wellbeing toolbox. Nonetheless, SNA has potential within animal welfare assessments that consider all aspects of an animal’s life, including their social role [3], as well as in assessing the impacts of moving animals in or out of an existing social group ahead of time [10].

Often, limited sample sizes preclude much zoo-based research concerning husbandry and management evidence from being published in the peer-reviewed literature [39]. Consequently, there is a lack of available case studies concerning the applications of SNA for individual groups of zoo animals, which are often limited in size, and many studies offer much broader explorations of the structure of animal sociality on a species level. Animal care staff are primarily concerned with the care and wellbeing of individuals [3,24,40], including their temperaments and how traits these interact with those of other members of their social group [41,42,43]. Case studies exploring how tools such as SNA can be applied to groups of individuals, even when sample sizes are limited, are invaluable resources for putting evidence into practice and reducing the traditional over-reliance on a “top-down” or generalised species-based approach founded in traditions and actions that have always been carried out in the past [39,44]. “Bottom-up” animal care approaches are instead founded on a relatively in-depth and holistic understanding of the individuals in human care, considering not only their species but also many other factors, including their individual life histories, temperaments, the context they are living in, and, indeed, their social lives. The proposal would, thus, be to utilise SNA as part of holistic animal wellbeing programmes to gain an in-depth understanding of the individual social experiences of animals, regardless of the size of the group with which we are working.

In the present article, a case study is presented alongside a discussion of the applications of SNA to provide an example of how SNA can be used to benefit the management of individual social groups through an understanding of the complexities of the social dynamics involved in individual groups. This study was conducted on a family group of four black lemurs (*Eulemur macaco*), a social species of cathemeral primate that, in zoos, is commonly housed in family groups consisting of a breeding pair and their offspring. Within these groups, complex patterns of female-dominated social hierarchies are present, with impacts on individual and group wellbeing [15,16,17]. The purpose of this study was to identify and examine social stressors after concerns were raised regarding the wellbeing of one individual in the group, who was the frequent target of aggressive behaviours. Social network analysis was used to quantify key metrics of social dynamics, including degree centrality, centralisation, and reciprocity, and examine the social role of each group member with regard to these metrics in order to illustrate the practical use of SNA. The methodology holds promise for wider application in zoos, aquariums, sanctuaries, and other facilities holding wild animals in human care.

## 2. Materials and Methods

### 2.1. Study Subjects

The present case study was conducted at Paradise Wildlife Park (PWP) in Hertfordshire, the United Kingdom, over a two-month (August–October) period in 2018. At the time of the study, the zoo housed a family group of 1.1.2 (male.female.juvenile) black lemurs (Table 1). The adult male “Bolek” was transferred to PWP from Opele Zoo, Poland, in 2013, and was the sire of the two juveniles involved in this study. The adult female and dam of the juveniles was born on-site at PWP. The juvenile male was no longer dependent on his mother at the time of the study; however, while she was weaned, the juvenile female was still partially dependent.

The group was housed in two alternative habitats throughout the study, transferring to the second habitat halfway through data collection stage. The first habitat, namely “Habitat A”, was adjacent to one other habitat, which housed black-and-white ruffed lemurs. The second habitat, namely “Habitat B”, was the same size as Habitat A (W 3.54 m, L 9.21 m, H 5.24 m), but we instead placed the group between two habitats containing different species of ruffed lemur. The move was unrelated to the management of the black lemurs and to the study itself and was instead related to reducing the prevalence of interspecific aggression between the ruffed lemur habitats. This was accomplished by moving the black lemurs into the existing black-and-white ruffed lemur habitat while moving the latter into the black lemurs’ previous habitat. Excluding the specific arrangement of enrichment, such as climbing frames, all other provisions in the two habitats were identical.

Both habitats included indoor housing, which was always open and available to the lemurs across 24 h days. There were additional shelters at the rear of both habitats with heat lamps that could be switched on during cold weather conditions. Habitats incorporated multiple and dynamic furnishings to provide varying levels, including logs, hammocks, ropes, and shelves. Both habitats had a single feeding table where a bowl of water was provided, and some food would be placed during feeding. Other feeding instalments included buckets attached to the side of the wire barrier. 

The animals were fed three times a day at the time of the study. The first feed occurred before the zoo opened at 08.00 and consisted of 360 g of Mazuri exotic leaf eater pellet with 90 g soaked for Bolek and scattered separately from the rest of the food items. The second feed took place around 13.00, and the third took place at around 15.00, with both feeds consisting primarily of a range of leafy, root, and other vegetables. Categories of vegetables fed were highly variable and, thus, not studied as a factor. Feed times did not occur at the same time every day and were variable depending on caregiver routines, but they generally fell within the hours described.

### 2.2. Behavioural Observations

An ethogram of social behaviours within the measured categories (Table 2) was devised and refined following a brief pilot study to ensure behaviours were relevant, as well as to ensure that intricacies between similar behaviours, such as aggressive versus playful chasing, could be identified.

Behavioural observations took place across ten total days, with three behavioural observation periods being used. As feeding has been associated with increased aggression in previous studies of animal behaviour [11,12,13,14], multiple time frames were selected to ensure social behaviour was not studied in isolation with respect to the fact that aggression may be triggered by husbandry events and inconsistent throughout the day. Each observation period was 1 h, with a total of 30 h of observation. The three observation periods took place on every day of study at 10:30–11:30 (no feed), 12:30–13:30 (noon feed), and 14:30–15:30 (afternoon feed).

Data were collected using a continuous event sampling method, wherein every instance of behaviour within each set of behavioural categories outlined via the ethogram was recorded. Data points were recorded according to which animal instigated the behaviour (the actor) and towards which animal the behaviour was directed (the reactor). Only the initiation of the behaviour was recorded; for example, if animals were engaged in huddling behaviour for ten minutes, only the moment that they started to huddle was recorded. 

Behaviours were recorded by an independent student observer who did not work with the study group directly and was not informed which individuals were involved in aggressive behaviour prior to the research. Video recordings were taken during the pilot study to allow the primary researcher to compare and confirm behavioural observations with those who possess experience working with the species. Video recordings were not used during the study itself due to logistical issues surrounding the positioning of the cameras and safety in an area that is open to the public. Direct observations were recorded using a pre-prepared observation sheet.

### 2.3. Social Network Analysis

Social network analyses were conducted via UCINET and NetDraw [45]. These software were selected due to their ease of use and applicability to a limited sample size, as well as the author’s familiarity with the program over other similar technologies and availability of training resources.

Utilised descriptive analysis indices included reciprocity, centralisation, and degree centrality. Reciprocity was used to determine whether interaction rates between individuals were mutually linked and, due to the limited sample size, measured using arc reciprocity. Reciprocity was measured on a scale of 0 to 1, with a result closer to 0 indicating “anti-reciprocity”, or a lack of reciprocal ties, while a result closer to 1 indicated “perfect reciprocity”.

Group centralisation was measured as a proportion on a scale of 0 to 1. Centralisation is an indicator of whether or not a group contains individuals who are more important than others, with a score closer to 0 indicating that social interactions are more evenly distributed, while a score closer to 1 indicates that there is a hyper-central individual with more importance. Degree centrality can then be used to identify which nodes hold more importance based on the number of connections that they hold with other individuals within the network. The “out” degree centrality indicates the extent to which a node initiates a behaviour towards another, while the “in” degree centrality refers to the extent to which a node is the recipient of a behaviour. In the present case study, degree centrality was indicated by the total number of interactions initiated or received compared to the network average for that behaviour. 

As the primary focus of this study was on examining the network structure in terms of describing the interactions between individual actors, rather than making inferences or testing hypotheses regarding the prediction of behaviour, further statistical analysis of the significance of SNA data was not required. 

### 2.4. Statistical Analysis

Prior to analysis, the data were checked with Shapiro–Wilk tests to determine the normality of variances. The variables analysed were the rate of the different social behaviours, whether animals were fed during that observation, and the habitat. Since the data were not normally distributed, non-parametric tests were deemed appropriate. The associations between feeding and the instance of different social behaviours were analysed using Kruskal–Wallis one-way ANOVA tests. To compare individual conditions, Mann–Whitney post hoc comparisons were separately conducted between each category. All data analyses excluding SNA were carried out utilising GENSTAT [46].

## 3. Results

### 3.1. Impact of Enclosure Change on Social Behaviour

As the enclosure changed partway through this study, it was not possible to gather enough data to carry out significant analysis of the impact that the enclosure change may have had on social behaviour. Despite this, preliminary comparisons were made in the total behaviour counts for each day of the study for both enclosures (Figure 1). Based on the limited amount of data collected, there was no significant difference in the instances of aggression, aversion, or affiliation behaviours between the two enclosures when tested using Mann–Whitney U comparison tests. While Figure 1 illustrates the day-to-day variability between the occurrence of behaviours, more data would be necessary to make a complete comparison between the enclosures. Figure 1 does, however, indicate that affiliative behaviours were the most frequent behaviours observed in the group, followed by aversion.

### 3.2. Impact of Feeding on the Rates of Social Behaviour

Statistical analysis of the data demonstrated that agonistic behaviour was variably impacted by feeding. Intragroup aggression levels (Figure 2) were significantly affected by feeding (Kruskal–Wallis one-way ANOVA, H = 6.85, df = 2, *p* < 0.05). There was significantly more aggression during both the noon and afternoon feeds compared to when there was no feed (*p* < 0.05). However, there was no significant difference between the noon and afternoon feeds.

The rates of aversive behaviours were similarly affected by feeding (Kruskal–Wallis one-way ANOVA, H = 10.59, df = 2, *p* < 0.05). There were significantly more aversive behaviours during feeding compared to the no-feed condition (*p* < 0.05), with no significant difference between the two feeding times. 

The impacts of feeding on affiliative behaviours were significant (Kruskal–Wallis one-way ANOVA, H = 8.51, df = 2, *p* < 0.05). Post hoc comparisons indicated that there was no significant difference between the no-feed condition and the noon feed; however, affiliation was significantly higher during the afternoon feed compared to both the no-feed and noon feed conditions (*p* < 0.05).

### 3.3. Social Network Analysis

SNA for aggressive interactions (Figure 3) indicated reciprocity at a level of r = 0.571, indicating that 57.1% of all aggressive interactions were reciprocated. The centralisation of the aggression network was reported at C = 0.43. Degree centrality results indicated that the adult female was more aggressive than other members of the group, initiating 141 aggressive behaviours compared to the network mean of 59.00 ± 57.953 s.d. The adult male was the recipient of the majority of aggressive interactions, receiving 179 interactions in contrast to the network mean of 59.00 ± 69.996 s.d. The juvenile male initiated the second highest (86) number of aggressive behaviours, compared to the adult male (9) and juvenile female (0).

SNA for aversive interactions indicated reciprocity at a level of r = 0.286. Centralisation was reported at C = 0.53. The adult male displayed the highest number of aversive behaviours, with 508 interactions recorded over the network mean of 146.75 ± 209.408 s.d. The adult female (328) and juvenile male (216) were the two individuals most likely to be the recipients of aversive behaviours (characterised by the other individuals avoiding or fleeing from them), compared to the network mean of 146.80 ± 132.240 s.d. Notably, the adult female was never aversive towards the other members of the group, and the adult male was never avoided.

SNA for affiliative behaviours indicated perfect reciprocity at a level of r = 1. The centralisation of this network was reported at C = 0.41. The network average for initiating affiliate behaviours was 427.50 ± 198.008, with the youngest female (670) initiating the most behaviours, followed by the juvenile male (524) and the adult female (383). The adult male was involved in fewer affiliative interactions compared to the rest of the group, initiating only 133 behaviours compared to the mean. Furthermore, he was the recipient of 171 affiliative interactions compared to the network mean of 427.50 ± 154.694 s.d.

### 3.4. Follow-Up Records

Anecdotal information was collected from caregivers involved in the care of the lemurs and extracted from ZIMS behavioural records following the study. Six months after the conclusion of data collection in March 2019, the adult male “Bolek” was separated from the group following increased bouts of chasing from the rest of the group and a physical altercation. Attempts to reintroduce the adult male to the juvenile male were unsuccessful. The juvenile male was transferred out of the zoo in December 2019.

Over the next seven months, several attempts to introduce the adult male back tfemales were unsuccessful due to physical aggression from the adult female. The juvenile female was transferred out of the zoo in July 2020, after which point the adult male and adult female were successfully reintroduced without issues. As of August 2023, the pair have remained together since their reintroduction.

## 4. Discussion

### 4.1. Case Study Analysis

The present case study demonstrates the value of social network analysis as a practical tool for managing social groups living in human care and analysing the social roles of individuals. The findings of the present case study show how SNA could be used to identify causes of and solutions for social instability after caregivers raised concerns surrounding aggressive behaviours within the black lemur group. It is worth highlighting that the observer in this case study, while not aware of which individuals were involved in the aggressive behaviours described, had been made aware of aggressive behaviours occurring prior to the observations. Awareness of these behaviours occurring may have inadvertently contributed to some level of bias in the recorded scores, as vigilance towards expected behaviours may be present. Should SNA be implemented by animal caregivers on a wider scale, it is worth both highlighting and acknowledging that observer familiarity with the animals, in addition to an awareness of concerns that have been raised surrounding behaviours, may lead to observer bias within the recorded results. 

Nonetheless, the social roles of all four individuals were consistent across SNA for all three behavioural categories following behavioural observations. The adult female was the most central individual across all networks, with a dominance style characterised by aggression towards the males, congruent with what would be expected of wild conspecifics [14,47]. The strongest social ties were observed between the two females, which was expected given that the juvenile female was still dependent [48]. Dyads that interacted prosocially more often displayed lower rates of aggression within those dyads, supporting previous observations that affiliative behaviours improve group cohesion and reduce the likelihood of aggression [49,50,51]. While patterns of female dominance identified were consistent with the current understanding of black lemur social behaviour [15,16,17], SNA highlighted that the adult male was the recipient of the majority of aggressive behaviours, frequently avoided the adult female and juvenile male, and had weak affiliative bonds with the social group. Understanding that positive and stable social environments are essential to positive wellbeing [1,3], the social environment and subsequent isolation from the rest of the group may be a potential cause of compromised welfare for this individual. 

Notably, rates of aversion were higher than rates of aggression, which indicates that aversion occurred in the absence of any aggressive instigation. Individuals who were targets of aggression, particularly the adult male, demonstrated avoidance and social submission towards those who were aggressive towards them. Aversion is a useful measure of social fear and submission [52], being a learned response to past experiences of aggression and social defeat [53,54]. Consequently, it could be inferred that these high rates of aversion were a consequence of negative individual perceptions regarding the aggressors, who may be perceived by those individuals as threats even in the absence of aggressive behaviours. Persistent wariness toward aggressors despite a lack of instigation may, thus, be a source of stress and compromised wellbeing for those individuals demonstrating higher rates of aversive behaviours.

It was not possible to gather enough data to carry out a significant analysis of the impact that habitat change may have had on social behaviours and, thus, what impact the change may have had on social wellbeing. Preliminary comparisons made based on the limited data collected indicated that there were no significant differences in the instances of aggression, aversion, or affiliation due to the move. As the new habitat was within the same facility, possessed the same dimensions, and was directly adjacent to the original, it is unlikely that this change would have impacted the results in a meaningful way.

### 4.2. SNA as a Tool for Finding Solutions

The analysis of behavioural rates indicated that the presence of food played an important role in catalysing bouts of aggressive behaviour. As observations surrounding non-feeding times occurred in the morning, it is also possible that the time of day was a factor. However, as black lemurs are cathemeral and, thus, irregularly active across a 24-h period [48], establishing cause and effect between whether animals are active because of feeding times or vice versa is challenging. Evidence from the present study and similar studies highlight the important role of food in motivating agonistic behaviour [11,12,13]. More dominant individuals, primarily the adult female and the male juvenile, may have been utilising aggression at these times to exert dominance over the adult male to increase their access to preferred food resources, or a combination of these factors may have led to an overall benefit of individual fitness. As specific feeding times, including length of time spent feeding, were not recorded, it is not possible from these results to determine the timing of aggression in relation to the specific feeding events or for how long aggression persisted following feeding. Therefore, understanding whether agonistic behaviours are confined to when food is present is difficult, though the observation of a low level of aversive behaviours, particularly in the absence of food, might indicate longer-term social stressors.

While some aggression is a fundamental aspect of sociality and stable social group dynamics [10,55], aggression becomes problematic when it is consistently to the detriment of individual wellbeing over prolonged periods, including when no specific aggressive behaviours are occurring. Targeted aggression may also put individuals at risk of poorer outcomes, such as receiving wounding during aggressive bouts. 

Solutions aimed at reducing aggression could be focused on feeding times, such as through alternative feeding presentations and strategies. Solutions could include wider food distribution within the habitat, which would provide the choice for animals to eat together or apart and reduce the likelihood of conflict [56]. Providing food in whole form instead of chopped has also been found to reduce aggression in several species and could be an additional consideration [57,58,59], as could presenting food in a manner that mimics natural foraging behaviours such as through enrichment [3]. Positive reinforcement training has also been used to promote cooperative feeding and reduce aggression in other lemur species, including ring-tailed lemurs (*Lemur catta*) and black-and-white ruffed lemurs (*Varecia variegata*) [60,61]. Future case studies could utilise a longitudinal approach to assess whether these strategies are useful interventions in similar cases by assessing changes to networks before and following implementation, using observed behaviour as a guide.

SNA could also guide housing decisions, such as determining when and which individuals may need to move on from their current social group or if separation is necessary to safeguard wellbeing. Following this study, the adult male lemur required separation from the rest of the group due to a severe instance of targeted aggression from the adult female. Based on an assessment of the SNA, one option for improving the social wellbeing of the adult male could have been to move the juvenile male to a new social group at another facility. The juvenile male, coming towards the age of sexual maturity, may have shown increased levels of aggression towards the adult male to bolster his social position within the group. The juvenile male was less central to the networks in comparison to the primary aggressor, the adult female, and so moving this individual would have a lesser impact on the overall social structure compared to moving the female. There was also the potential for the presence of a dependent juvenile to be a catalyst for increased aggression from the adult female, as she may have perceived the adult male as a threat to her access to resources. Aggression has been observed to increase around the time of birth in wild lemurs [46,62], which may be an alternative explanation for the social dynamics observed in the case study.

The younger male was eventually moved to another facility; anecdotal post-intervention observations suggest that these changes to the social group may have been effective at alleviating aggression and increasing affiliation for some time, though it was only once both juveniles had left the group that the successful reintroduction of the two adults was possible. To this end, SNA may aid in planning for conservation breeding programs, such as the black lemur EEP, by identifying the ideal age for offspring to move on to new social groups before social issues begin to occur. Longer-term monitoring via SNA would be beneficial to confirming any lasting improvements in social position and wellbeing as a result of these changes and monitoring for early indicators of compromised social wellbeing in the future. 

### 4.3. SNA in Holistic Wellbeing Assessments

While a small study, the present case demonstrates how network analyses of social groups including as few as four individuals can provide empirical insights that can be used to the benefit of practical management strategies and animal wellbeing. Holistic animal wellbeing assessments should incorporate measurements of social wellbeing alongside all other aspects of their physical and psychological health, with an understanding that the social environment can have profound impacts on the mental states of animals alongside all other elements of their lives [1]. Towards this goal, SNA can reveal intricate and subtle relationships between dyads and within groups that traditional measurements of social behaviour rates would not otherwise identify. SNA has valuable potential as part of evidence-based assessments of wellbeing alongside other tools that assess other areas of the animal’s lived experience, following frameworks such as the Five Domains [1]. For example, SNA can provide empirical evidence as to how well an individual is integrated into their social group, the rate of positive versus negative interactions, and the impact of social hierarchies on wellbeing. Paradise Wildlife Park, as a practical example, conducts a monthly assessment of geriatric animals, which includes a social interactions component. The completion of this assessment requires evidence regarding the types and frequencies of social interactions for which SNA could form the basis, contributing to an evidence-based “snapshot” of the social environment. Overall, the potential of SNA as a proactive tool in the prediction, monitoring, and mitigation of wellbeing concerns warrants further investigation.

Collecting longitudinal SNA data has broad applications for future research into the social behaviour of many species. Retroactive studies of SNA data may examine the factors that influence social changes over time, for example, and the impacts of various husbandry events on social structures can also be assessed. An understanding of species- and individual-level responses to changing group dynamics is broadly useful for future animal care decisions, such as the planning, monitoring, and evaluation of introductions and translocations with respect to the social lives of animals. Continued monitoring could reveal subtle long-term effects on individual relationships [35]. 

Longitudinal SNA will also facilitate the observation and identification of natural fluctuations in social roles and relationships unrelated to husbandry. For example, there may have been an increasingly dominant role of the juvenile male as he approached sexual maturity, as demonstrated by his aggression towards the adult male. Such interactions may reflect the natural life history of the species, though these effects would be better understood via SNA conducted throughout the animal’s lifespan. Ongoing SNA could further aid proactive welfare assessments and the management of animals by monitoring for early signs that negative social pressures may be intensifying, such as the declining social position of the adult male, and facilitate timely interventions. SNA could also evaluate the impacts of future interventions aimed at improving welfare. Comparing networks before and after implementing such changes would reveal if these actions successfully altered social dynamics and improved social wellbeing as intended. More broadly, case studies applying SNA to a diverse range of groups across populations contribute valuable insights to inform best practices and compassionate care of animals in general.

It must be recognised that there are obstacles to implementing SNA on a wider systematic scale. A straightforward method to complete, time, personnel, and expertise is nonetheless required to collect and collate the data, as is relevant training and guidance regarding planning and conducting data collection as well as analysis of the data itself. While the data may be straightforward to collect, a thorough understanding of the social behaviour of the species, as well as of analysing data itself, is required for any assessment tool to be meaningful. Furthermore, many smaller zoos may lack the time and resources to conduct social network analyses on a continuous and systematic scale. There is, often, a research bias towards zoos with larger sample sizes [36], which can leave smaller zoos without the resources they might need to fully utilise tools such as SNA. Care staff often lack the time to collect extensive data for research between animal care and other responsibilities [3], and the collection of sufficient data for meaningful SNA is time-consuming. External researchers, including collaboration with colleges and universities, are key to facilitating further study. 

While sample sizes preclude broad generalisations, limited population numbers should not dissuade researchers from conducting SNA case studies. Such studies aid in improving our understanding of species-typical social structures and individual variations, which may aid care decisions for related taxa in other collections and improve individual and group wellbeing. By demonstrating SNA as a practical, evidence-based tool even for small groups, this case study supports the increased application of SNA and contributes to the advancing multidisciplinary approach to optimising individual- and group-level wellbeing.

## 5. Conclusions

The presented case study demonstrated that even with limited scope, social network analysis can reveal useful insights into the social dynamics of wild animals living in human care. As such, this case study is a useful demonstration of how SNA can be used on an individual group scale within zoos and aquariums. Mapping the affiliative and agonistic interactions between group members identified differences in social roles, including a subordinate individual receiving high aggression and potentially experiencing compromised wellbeing due to his social position. Anecdotal follow-up observations demonstrated that removing an individual identified as a possible cause of social tension had some success in temporarily improving the social status of the targeted individual. 

Though this is a preliminary study, SNA as a complementary tool in monitoring and managing the complex social lives of animals and in informing management strategies is showcased. Larger studies across diverse taxa are needed to confirm the generalisability and optimise methodologies, as are longitudinal studies, which track changes in social networks over time and in response to changes in housing or husbandry. Overall, despite having limitations when applied to small social groups, SNA holds great promise as part of a systematic, proactive, and holistic wellbeing management toolbox, presenting a prospect for fundamentally improving how zoos and aquariums monitor and manage the social lives of the animals in their care. 

## Figures and Tables

**Figure 1 animals-13-03501-f001:**
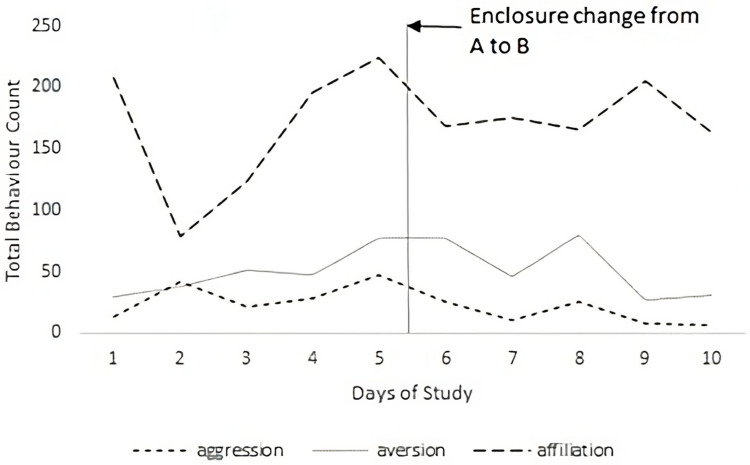
Total counts for each behaviour for the whole day (aggression, aversion, and affiliation) over the 10 separate sample days during the 2-month study period, with the point of enclosure change marked by a vertical line.

**Figure 2 animals-13-03501-f002:**
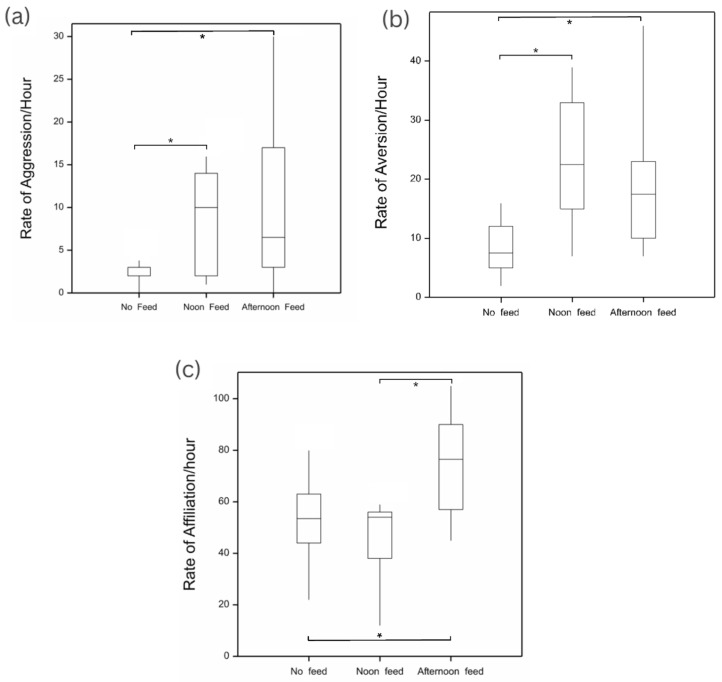
The relationships between feeding times and the rates of (**a**) aggressive, (**b**) aversive, and (**c**) affiliative behaviours (expressed as behaviours/hour). Significant differences between pairs are denoted with an asterisk (*).

**Figure 3 animals-13-03501-f003:**
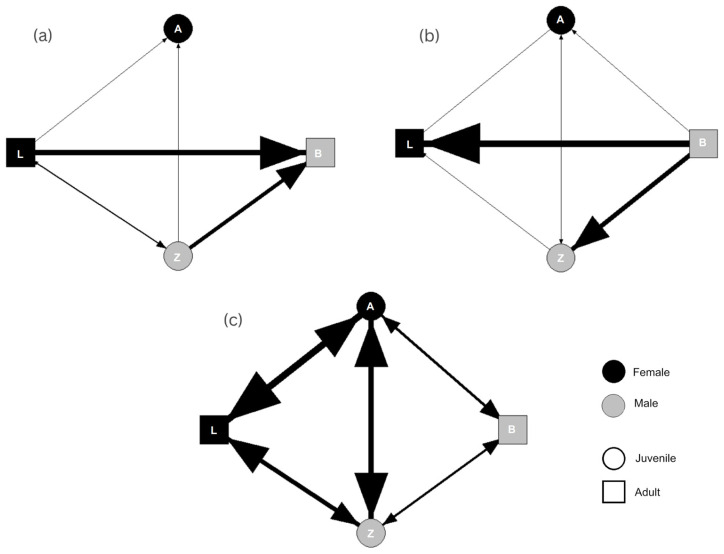
Sociogram for total (**a**) aggressive, (**b**) aversive, and (**c**) affiliative interactions witnessed in a family group (1.1.2) of black lemurs. Line thickness is proportional to the number of interactions observed between dyads. Letters represent the animal ID. The direction of arrows indicates the initiator vs. receiver of behaviour.

**Table 1 animals-13-03501-t001:** Name, age, and sex of black lemurs housed at Paradise Wildlife Park at the time of this study.

Subject ID [Initial]	Age during Study	Sex
Bolek (“Bo”) [B]	8 years, 7 months	Male
Louise [L]	13 years, 4 months	Female
Zafy [Z]	1 year, 5 months	Male
Adala [A]	4 months	Female

**Table 2 animals-13-03501-t002:** Ethogram of social behaviours displayed by black lemurs.

Behaviour Category	Behaviour	Description
Affiliative	Social rest	Two or more animals sitting inactive in close or full contact.
	Greet	Briefly sniffing or nuzzling another animal.
	Groom	Grooming of any body part of another animal using hands, teeth, or tongue.
	Play	Reciprocal chasing with no apparent goal or function besides recreation.
Agonistic—Aggresive	Pounce	Jumping towards another animal.
	Chase	Running pursuit of another animal.
	Lunge	Thrusting of body towards another animal.
	Cuff	Striking of another animal with a forehand.
	Bite	Oral seizure of another animal’s fur or limb(s).
Agonistic—Aversive	Displaced	Abandonment of current location due to the approach of another animal.
	Flee	Rapid locomotion away from another animal.
	Cower/flinch	Pulling of body away from another animal.
	Back away	Walking backwards away from another animal.
	Jump away	Single jump away from another animal.

## Data Availability

All data relevant to the study are included in this article.

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
