# Peer review of "Social Network Analysis as a Tool in the Care and Wellbeing of Zoo Animals: A Case Study of a Family Group of Black Lemurs (Eulemur macaco)"

_animals, 2023, doi:10.3390/ani13223501_

Round 1
Reviewer 1 Report
Comments and Suggestions for Authors
I find the research article well written, structured, and justified in its aims and methodologies implemented.
I have the following suggestions:
1 - In the simple summary, in the abstract, and in the introduction, instead of talking of a "small group" I would suggest providing the actual number of animals. Depending on the species, settings, etc. small or large are relative descriptions.
2 - The abstract could benefit from a sentence or two at the end regarding the conclusion and implications of the study.
3 - In the introduction, line 46, The Five Domains Model should be explained briefly.
4 - Hyperlink at reference 2, appearing first at line 50, does not work.
5 - Table 1 contains a column regarding date and place of birth. I suggest dropping this information from the table as it is redundant (age is already contained in the next column) and not relevant for the table itself. I could imagine adding to the table actual SNA values for each animal and each social domain considered.
6 - Table 2 is difficult to parse in its current format. As I do not have a specific suggestion I will just say that the bold text could be an additional column to the right, rather than title rows.
7 - Methodological description of the data acquisition protocol could be transformed into a methodological figure, where also the animal housing is presented in a schematic way (the two habitats are of the same size) and the feeding and the observations can be compared visually if illustrated in a line from morning to evening.
8 - Section 3.2 should include more details regarding how, in what format, by whom, the data was acquired / recorded.
9 - Section 3.4., line 201, the choice to why UCINET and NetDraw were used should be discussed briefly. Have other tools been used?
10 - Figures 1, 2, and 3 could be compiled into a single figure, as individual panels. Also raw data should be visible as individual points on top of the boxplots, preferibly split by animal so that it can be judged whether the overall trend depends on specific animals (as it is suggested by SNA analysis)
11 - Figures 1, 2, and 3: the meaning of "A" and "B" it is unclear. In the SNA there are also A and B and those are animals (Adala and Bo). I suggest adopting the classical visualization of lines and stars to indicate significance and / or avoid potential confusion with animals' labels
12 - Figures 4, 5, and 6 can similarly be put as panels of the same figure so that differences between bhaviors can be better evaluated.
13 - It is unclear to me whether a statistical evaluation of the significance of the SNA results was carried. It is possibile that the SNA does not require a statistical evaluation, as results are descriptive. That would be fine, but in either cases this needs to be discussed as naive readers might not be aware.
14 - Discussion, lines 312-314: please make clear at this stage that the SNA approach described in this study was triggered by concerns of abnormally high aggression specifically from one individual to another. This is very important and somewhat concerning for the study (see 15)
15 - As the SNA was triggered by concerns of aggression in the group, it has to be clear whether whoever scored the ethograms was aware of such concern or not. While the SNA results seem to be strong, potential scorer biases can not be neglected and should be, at the very least, highlighted throughout the text
16 - I personally think that in discussion, line 331-335 could be stressed more and given more space.
I have no more comments. I would like to thank the authors for the stimulating and well written description of their work.
Author Response
REVIEWER 1:
Comments and Suggestions for Authors
I find the research article well written, structured, and justified in its aims and methodologies implemented.
I have the following suggestions:
1 - In the simple summary, in the abstract, and in the introduction, instead of talking of a "small group" I would suggest providing the actual number of animals. Depending on the species, settings, etc. small or large are relative descriptions.
Thank you for your suggestion. You raise a good point regarding the relativity of descriptors of group size; arguments could be made in the context of captive animals any group is ‘small’ (and for black lemurs specifically this group size is probably within the average). The simple summary, abstract, and introduction have been amended to specifically refer to the number of individuals for clarity. Throughout the manuscript, references to “small groups” where not specifically referring to a named group have been amended to refer to “limited sample sizes” instead.
2 - The abstract could benefit from a sentence or two at the end regarding the conclusion and implications of the study.
Thank you for the suggestion. We hope that amendments made to the abstract, in combination with a concluding sentence, have been made to summarise the conclusion and implications more clearly. A clearer focus on the applications of SNA have been made in response to comments from another reviewer, and so we hope the focus of the paper and its practical outcomes is more explicit in the amended versions.
3 - In the introduction, line 46, The Five Domains Model should be explained briefly.
Thank you for your suggestion. A brief introduction to the Five Domains Model has been included to highlight the importance of the framework more clearly, as well as more clearly linking this model to the intentions of the study.
4 - Hyperlink at reference 2, appearing first at line 50, does not work.
Thank you for highlighting this issue – the link has been amended to a working URL.
5 - Table 1 contains a column regarding date and place of birth. I suggest dropping this information from the table as it is redundant (age is already contained in the next column) and not relevant for the table itself. I could imagine adding to the table actual SNA values for each animal and each social domain considered.
The date and place of birth column has been removed as per your suggestion, thank you. Upon reflection, information regarding the origin of the animals is already described in the text and the date of birth is indeed redundant in conjunction with age and we are happy to remove this information from the table.
As the SNA values fall under the results and are described there, we have elected not to include this information in this table. However, we have added the initial ‘code’ used for each animal to the table; though this information is presented in the figures themselves as well, we hope this addition improves clarity and makes the figures themselves easier to parse later in the manuscript.
6 - Table 2 is difficult to parse in its current format. As I do not have a specific suggestion I will just say that the bold text could be an additional column to the right, rather than title rows.
Thank you for highlighting the difficulties in the original table. Upon review we would agree the table is challenging to read. The table has been amended according to your suggestion with a new additional column denoting the behavioural category sub-sections as opposed to using title rows; hopefully, the table is easier to parse in the amended format.
7 - Methodological description of the data acquisition protocol could be transformed into a methodological figure, where also the animal housing is presented in a schematic way (the two habitats are of the same size) and the feeding and the observations can be compared visually if illustrated in a line from morning to evening.
Thank you for your suggestions. As the feeding time was not consistent daily, we have concluded that a visual depiction of the feeding times versus the observation times may potentially be misleading. We would not want to imply that food was only available within that window, or that feeding behaviours only occurred in those times. As such it has been elected not to include a visual representation of the timings, though we thank you for the suggestion.
8 - Section 3.2 should include more details regarding how, in what format, by whom, the data was acquired/recorded.
Thank you for highlighting this missing information. Additional information has been included on the observer and their level of familiarity with the animals, as well as some information that you requested in comment 14/15 regarding whether that observer was aware of aggression prior to the study.
9 - Section 3.4., line 201, the choice to why UCINET and NetDraw were used should be discussed briefly. Have other tools been used?
Thank you for your suggestion. A brief explanation has been added to elaborate on why UCINET and NetDraw were chosen for this study: “These software were selected due to their ease of use and applicability for a small sample size, as well as the author’s familiarity with the program over other similar technologies and availability of training resources.”
10 - Figures 1, 2, and 3 could be compiled into a single figure, as individual panels. Also raw data should be visible as individual points on top of the boxplots, preferibly split by animal so that it can be judged whether the overall trend depends on specific animals (as it is suggested by SNA analysis)
Thank you for your feedback. Please see comment below.
11 - Figures 1, 2, and 3: the meaning of "A" and "B" it is unclear. In the SNA there are also A and B and those are animals (Adala and Bo). I suggest adopting the classical visualization of lines and stars to indicate significance and / or avoid potential confusion with animals' labels
Thank you for your suggestions. To address both comments, we have compiled the three figures into one with different panels denoted by (a), (b), and (c). We have also adopted the classical visualisation of lines and stars to indicate significance.
12 - Figures 4, 5, and 6 can similarly be put as panels of the same figure so that differences between behaviors can be better evaluated.
Thank you for your suggestion. Figures 4, 5, and 6 have been consolidated into a new Figure with individual panels labelled (a), (b), and (c) similar to the previous suggestion. We agree that this format allows for much more straightforward comparison of the results.
13 - It is unclear to me whether a statistical evaluation of the significance of the SNA results was carried. It is possibile that the SNA does not require a statistical evaluation, as results are descriptive. That would be fine, but in either cases this needs to be discussed as naive readers might not be aware
Thank you for highlighting the lack of clarity here. This has now been addressed in the materials and methods which now refer to descriptive analysis of SNA data specifically, as well as the addition of a specific explanation as to why statistical evaluation was not required for the SNA: “As the primary focus of this study was on examining the network structure in terms of describing the interactions between individual actors, and not on making inferences or testing hypotheses regarding predicting behaviour, further statistical analysis of the significance of SNA data was not required.”
14 - Discussion, lines 312-314: please make clear at this stage that the SNA approach described in this study was triggered by concerns of abnormally high aggression specifically from one individual to another. This is very important and somewhat concerning for the study (see 15)
15 - As the SNA was triggered by concerns of aggression in the group, it has to be clear whether whoever scored the ethograms was aware of such concern or not. While the SNA results seem to be strong, potential scorer biases can not be neglected and should be, at the very least, highlighted throughout the text
Thank you for highlighting this valid concern – we agree that this is important to mention and especially worth discussing in the context of this method being more widely applied by caregivers, who will be familiar with the animals and their behaviour. It is now, hopefully, clearer that the behaviours observed were a trigger for the study in the introduction: “The findings of the present case study demonstrated how SNA could be used to identify potential social instability after caregivers raised concerns surrounding aggressive behaviours within the group.” We have also amended the methods as discussed under point 8.
The observer in this study was made aware of aggressive behaviours that were occurring, but it was not highlighted to the observer who was instigating these behaviours and who was the target (or whether just two or multiple individuals were involved). This has now been included in the materials and methods (see 8) and reference has been made in the discussion as well: “It is worth highlighting that the observer in this case study, while not aware of which individuals were involved in the aggressive behaviours described by caregivers, had been made aware of aggressive behaviours occurring prior to the observations. Awareness of these behaviours occurring may have inadvertently contributed to some level of bias in the recorded scores, as in this and similar cases vigilance towards particular expected behaviours may be present. Should SNA be implemented by animal caregivers on a wider scale, it is worth both highlighting and acknowledging that observer familiarity with the animals, in addition to an awareness of particular concerns that have been raised surrounding behaviours, may lead to observer bias within the recorded results.”
We hope this amendment improves the clarity of the work and addresses this important concern appropriately.
16 - I personally think that in discussion, line 331-335 could be stressed more and given more space.
Thank you for this suggestion. These lines have been moved into their own paragraph to give them more room to breathe, and the points highlighted have been expanded upon with additional discussion to highlight the importance of aversion as an indicator more thoroughly.
I have no more comments. I would like to thank the authors for the stimulating and well written description of their work.
Reviewer 2 Report
Comments and Suggestions for Authors
The authors report on an easy applicable method to monitor social dynamics in animals living under human care. It is a case study with a very limited sample size, however it shows the value of network analyses as a tool for management decisions on group compositions and relocation of individuals. This is a well-written article, there are only a few things that need attention.
The sample size is rather restricted due to only 10 days of observations within a 2-month period. Why was the observation time restricted to only 1 hour? This might especially influence the observations since the feeding was not at fixed times
Please give some more detailed information on the occurrence of the aggressive/aversive behaviours around feeding times. Did the behaviour already start in anticipation of feeding before the food was given?
Please also indicate how long after the presentation of the food the aggressive behaviours occurred. Is that a longer-term stressor or a short-lived event?
The discussion is a bit too long and should be shortened
Comments on the Quality of English LanguageThis is a well written manuscript
Author Response
REVIEWER 2:
The authors report on an easy applicable method to monitor social dynamics in animals living under human care. It is a case study with a very limited sample size, however it shows the value of network analyses as a tool for management decisions on group compositions and relocation of individuals. This is a well-written article, there are only a few things that need attention.
Thank you, we really appreciate your feedback here.
The sample size is rather restricted due to only 10 days of observations within a 2-month period. Why was the observation time restricted to only 1 hour? This might especially influence the observations since the feeding was not at fixed times
Thank you for highlighting this point.
Please give some more detailed information on the occurrence of the aggressive/aversive behaviours around feeding times. Did the behaviour already start in anticipation of feeding before the food was given? Please also indicate how long after the presentation of the food the aggressive behaviours occurred. Is that a longer-term stressor or a short-lived event?
Thank you for raising these questions, which highlight an issue with the original methodology and data collection which had not previously been acknowledged within the manuscript. The specific daily timing of the feed was not recorded on the study days as it did not always occur within the hourly observations and was not officially recorded by the caregivers. Furthermore, as only social behaviours were recorded and feeding behaviours were not, correlations between feeding times and length of aggressive bouts cannot be made (e.g., feeding is not confined to the single event of food being provided, as the animals may have been eating throughout the hour-long observation period).
While it is not possible to answer these specific questions retrospectively, they are nonetheless important questions and warrant discussion. The presence of a low level of aversion in the absence of food is the primary indicator that social stress persisted outside of feeding times, in connection with the earlier analysis within the discussion of aversion as a learned response to social defeat. The following information has been added to the discussion in 4.2 in respect to this comment: “As specific feeding times, including length of time spent feeding, were not recorded, it is not possible from these results to determine the timing of aggression in relation to the specific feeding events or for how long aggression persisted following feeding. Therefore, understanding whether agonistic behaviours are confined to only when food is present is difficult to ascertain, though the observation of a low level of aversive behaviours particularly in the absence of food might indicate longer-term social stressors.”
The discussion is a bit too long and should be shortened
Thank you for this suggestion. This section has been reviewed for opportunities to be more concise, though it was necessary to balance this suggestion with comments from other reviewers requesting clarification on certain elements of the discussion. The amended discussion has been shortened wherever possible to do so, though to meet the requirements of all reviewers the length is not significantly reduced. We hope this is justifiable in the context of the additions made, which are worthwhile improvements to the manuscript in their own right.
Reviewer 3 Report
Comments and Suggestions for Authors
Author Response
REVIEWER 3:
The authors make a compelling case for the benefits of using social network analysis even on small social groups to provide an empirical basis for management. While keepers could see without formal analysis that one individual was the target of aggressive behavior, SNA provided deeper insights into the social dynamics of the group and led to the removal of the young male. Furthermore, the results of a feeding analysis indicated that feeding had a significant impact on behavior, which has many implications for husbandry practices. While continued study after management changes would have added significantly to the manuscript, the point holds that SNA is a valuable tool that warrants further investigation. Overall, the study is presented clearly, but filling in gaps in the methods in addition to cleaning up wording and organization is necessary before publication.
Major comments Info missing from methods: Was there one observer only? Were visitors present and was this accounted for as a potential influence on behavior?
Thank you for highlighting this missing information. More information on the observer has been included. The zoo was open at the time of the study and so visitors were present. Information on visitor density was not recorded and not measured as a potential influence. As a small zoo, visitor density did not vary significantly including between feeding times.
The methods should outline how data was treated including which variables were analyzed. No statement regarding initial data checks/data structure is included. Presumably these data were not normal and this is why the Kruskal Wallis/Mann Whitney was used? Please include this information.
Thank you for highlighting this missing information. Information on the Shapiro Wilk test used has been added to the Materials and Methods.
Lines 219-247: The presentation of feeding results is the first time in the manuscript that feeding is brought up as an item of interest. Food aggression should be a topic in the introduction and information regarding which comparisons were made should be included in the methods (for instance, which observations were included in the “no feed” category?) How do you know that it was feeding causing the differences in affiliative/aggressive/aversive behavior versus time of day?
Thank you, we have made amendments throughout the manuscript to ensure food aggression is introduced earlier in the introduction. We hope this improves the flow of information throughout the paper. The no-feed category referred to the morning observation and so, indeed, there may have been an impact of time of day. As black lemurs are cathemeral with irregular periods of activity it is difficult to establish cause and effect between the times they feed, the time they are awake, and the times they are engaging in different social behaviours. This has been made clearer in the discussion: “As observations surrounding non-feeding times occurred in the morning it is also pos-sible for time of day to have been a factor. However, as black lemurs are cathemeral and thus irregularly active across a 24-hour period [48] establishing cause and effect between whether animals are active because of feeding times or vice-versa is challeng-ing. Evidence from the present study and of similar studies highlight the important role of food in motivating agonistic behaviour [11-13].”
Lines 338-340: Unless methods for these “preliminary analyses” between habitat A and B are fully described, this statement should not be included. Since equal observations were taken from both habitats, it shouldn’t bias the results and should not be as big of a focus as it currently is within the manuscript.
Other reviewers requested elaboration on this information and further development; as such, there is now a greater focus in contrast that is now supported with the results. Of course, on further review, this may be information that is better off removed entirely. We opted to include this information for transparency and clarity of results at this stage.
The words “potential” occurs 29 times in the manuscript. Many statements can be made stronger and more concise.
Thank you, we have reviewed all instances of the word “Potential” in the manuscript and made relevant amendments wherever possible.
Minor comments Line 42: “many of the myriad” is awkward
Thank you, this statement has been amended and shortened.
Lines 46-50: hard to follow this lengthy sentence
Thank you, this sentence has been amended and split for clarity. We hope the updated version is easier to parse.
Line 51: awkward phrasing “hereafter referred to interchangeably with wellbeing”
Thank you, we have replaced this phrase with a single sentence: “For brevity, the terms welfare and wellbeing are used synonymously within this paper.” This distinction has been deemed important as not all papers do use the terms interchangeably, though a lengthy discussion on which term is most appropriate would be out of the scope of the article.
Lines 102-105: hard to follow this lengthy sentence
Thank you, we have split this sentence.
Line 149: not clear what the 1.1.2 refers to
Thank you, it is common notation in zoo animal inventory but indeed will be unclear to naïve readers. We have added a note to ensure it is clear that 1.1.2 refers to the number of males, females, and adolescents in the group.
Lines 155-162 (also 336-343): not clear why this move to habitat A to B was done. Was it an experimental move as part of the study or a management decision made independent of the study? Additionally, the two habitats were the same size, but were they identical in all other regards as well?
The move was unrelated to the husbandry of the black lemurs and made independently of the study. We had include a brief note that the move was due to interspecific aggression between ruffed lemur habitats; essentially the black lemurs swapped habitats with one of the ruffed lemur species to create a ‘buffer’ between the black-and-white and red ruffed lemurs. We have expanded on it a little more in the materials and methods: “The move was unrelated to the management of the black lemurs and to the study itself and was instead related to reducing the prevalence of interspecific aggression between the ruffed lemur habitats. This was accomplished by moving the black lemurs into the existing black-and-white ruffed lemur habitat while moving the latter into the black lemurs previous habitat. Excluding the specific arrangement of enrichment such as climbing frames, all other provisions in the two habitat were identical.”
Line 172: how was it ensured that only he ate this? I think you mean the pellets were scattered away from the other food items not scattered away from the group?
Thank you, you are correct in your assumption that the food was scattered separately from other food items. No specific other alterations were made to ensure the rest of the group did not eat the food items intended for Bolek.
Lines 185-187: Would be easier to follow if the three daily time periods were listed at the end of the sentence.
Thank you, we have amended this sentence and the paragraph as a whole to restructure.
Line 195: Change this heading to “Statistical analysis” and consider moving to the end of the methods section
Thank you for highlighting this error. We have made the recommended changes.
Line 259, 272, 285: Consider changing “edge weights” to “line thickness”
Thank you for the suggestion, we have made this amendment.
Lines 303-314: This paragraph fits more logically in section 4.2, but could be deleted altogether. Feels redundant.
Thank you for raising this point. To maintain the paper’s focus, which was a challenge highlighted by another reviewer, it seems pertinent to ensure that the case study is tied back to the uses of SNA early in the discussion. However, we agree much of this information is redundant and have significantly cut back on this paragraph to flow more seamlessly into the discussion of this section. We hope this flow is logical.
Lines 312-314: If already witnessed previously by caregivers, what does the SNA add? “Identifying soial instability” has already been witnessed by caregivers prior.
Thank you for raising this, we have made it clearer that SNA provides the depth needed for identifying causes and solutions as a data-driven decision-making tool.
Line 334: Used the line “could be inferred” twice in this paragraph. Consider changing wording.
Thank you, we have restructured these paragraphs to give more space and focus to the discussion of aversive behaviours per another reviewer’s suggestion. In this restructuring we have also amended the wording to avoid repetition.
Lines 354-355: I would counter that aggression is nearly always detrimental to individual well-being, so at what point does it truly become an issue?
You raise a very good point here, thank you for the comment. Indeed, aggression is always a detriment to the individual in the moment that it occurs, regardless of the possible beneficial impacts on social structure and maintaining group dynamics.
We have specified more clearly that problematic issues arise when this is consistent, prolonged, and impacts animal wellbeing even outside of the specific instances of aggression – hence the relevance of discussing aversive behaviours.
Line 355: From this line on, authors discuss potential solutions. Could be a separate paragraph to separate this large block of text.
Thank you, we have split up the paragraphs based on this suggestion.
Lines 373-375: The following paragraphs talk about how SNA can influence housing decisions in captivity. Make the topic sentence more closely align with the subject matter.
Thank you, we have amended the topic sentence for this paragraph: “SNA could additionally guide housing decisions, such as determining when, and which individuals, may need to move on from their current social group, or if separation is necessary to safeguard wellbeing”.
Lines 416-419: Sentence is hard to follow.
Figures 1-3: It strikes me as strange that all the figure captions include statements like A compared to B, B to B etc. instead of just naming those the actual feeding conditions.
Thank you for raising this concern. Based on feedback from other reviewers we have consolidated these graphs into one figure with panels and used standard lines-and-asterisks notation to indicate significance.
Figures 4-6: Adding the absolute number of behaviors in figures 4-6 next to each animal ID would be immensely helpful. If you don’t choose to do this, at the very least it needs to be added into the text for aggressive behavior the way it already is for affiliative and aversive. I would also consider writing out the animal name rather than using the letter. I didn’t put two and two together immediately.
Thank you, we have followed the suggestion to add the absolute number of aggressive behaviours in the text as we had for affiliative and aversive for consistency between the three behavioural categories. To tackle the latter issue, we included in the original table the letter we used for each animal.
Reviewer 4 Report
Comments and Suggestions for Authors
Thank you very much for the opportunity to evaluate this very interesting work. I think its results are valid and should be presented. However, I have made a series of comments throughout the text to improve the writing and in which I also ask you to explain some points better.

Author Response
REVIEWER:
Thank you very much for the opportunity to evaluate this very interesting work. I think its results are valid and should be presented. However, I have made a series of comments throughout the text to improve the writing and in which I also ask you to explain some points better.
Thank you for your extensive and useful feedback on the manuscript.
Some of the comments come down to disagreements in terminology and while arguments are valid this, to me, is an area of “agree to disagree” as opposed to a barrier to the quality of the work. I prefer to use the term wellbeing over welfare as I find it to be more inclusive of the physical and psychological states of animals, though the argument of which term is most appropriate could surely form a whole paper and would be tangential to explain more thoroughly in this manuscript. As the connection between the two terms is well-known and sufficiently introduced in the paper, I would argue that retaining the original wording is not problematic. The same could be said of replacing aversion with submission. And, lastly, I do not personally agree with using ‘it’ pronouns for animals and especially not when the sex of the animal is known, though I understand this is a matter of personal preference particularly in academic literature.
“1.1.2” denotes the distribution of sexually mature adults of each sex and juveniles (male.female.juvenile) and is the standard for communicating such information in zoo inventories and further has widespread usage in many papers studying zoo species where it is not often elaborated upon. It would be expected that the target readership of the article would know what this means and how to read it, though we have added a brief notation (1.1.2 (male.female.juvenile)) to hopefully communicate this information to naïve readers more clearly.
We have amended the results section and added an initial section (now 3.1) which showcases the results of the day-to-day behaviour rates and highlights the lack of significant difference between the habitat 1 and habitat 2 results, as highlighted in the discussion. Thank you for highlighting this missing information.
Reviewer 5 Report
Comments and Suggestions for Authors
The study proposal is very interesting, and this field of knowledge is little developed in zoo research. I have no doubt that it can contribute to the management of animals, particularly social animals. However, the introduction is not focused on the case study (Eulemur macaco) and the interesting topic: social behavior. In this section, there are a number of problems indicated that do not help the reader to understand the specific objective. The relationship between well-being and social network analysis is speculative. It is necessary to be more specific, considering that behavior is one of the domains of animal welfare. Therefore, its study does not guarantee that the other domains can be directly affected. The introduction seems like a draft of a review of the topic, rather than a specific context to understand why this study or case study deserves to be published. On the other hand, it highlights some difficulties of research in zoos (lines 112-113); and this is not true base on the large number of papers. Furthermore, I concern the assertion that certain methodologies for the study of behavior are easy to apply by non-entry-level people. Although it may be true that obtaining data does not require much training, the question that motivates its sampling, analysis and discussion is only possible with high training (lines 91-96). The introduction lacks specificity even in the specific objective, negatively affecting the interpretation of the aim of this research. This is another example of the lack of specificity. Furthermore, social stressors are mentioned when they were not previously clearly identified, nor associated with real problems of the study species. After reading M&M, the study is a behavioral investigation. So, I strongly suggest that the Authors focus on this issue, and consider the potential of observation and its link to animal welfare for the discussion. Furthermore, considering the recording rule (lines 185-188) and, for example, the first results presented (lines 219-225), I presume that the authors considered feeding as a possible stressor of the group's social relations. This is not developed, neither in the introduction nor justified in M&M. Finally, considering that social structure is founded on behavioral interactions among individuals, and this case study was focused on a family group, a better introduction is necessary to prepare the reader for understanding possible results and differences among studied individuals. This study did not employ similar observational units, there were adults and juveniles. Additionally, reading the description of the observations I cannot understand the recording methods. It appears that Authors applied a behavior sampling rule and a continuous recording rule. It would not be an instantaneous sampling. This should be reviewed and associated with the core question and/or the experimental design (why did Authors decide to sample before and after feeding?). I feel that a better description is needed adding information about: attributes of individuals, not only age but also role, dominance rank; types of relationships studied: bonds, dependence, dominance.
I consider that the Author´s approach is original for this species and zoo research, however, the lack of focus in the manuscript, the lack of clarity in the sampling rules and of justifications for the structure or sampling periods, do not allow me to accept this proposal to be published.
Author Response
REVIEWER 5:
The study proposal is very interesting, and this field of knowledge is little developed in zoo research. I have no doubt that it can contribute to the management of animals, particularly social animals. However, the introduction is not focused on the case study (Eulemur macaco) and the interesting topic: social behavior. In this section, there are a number of problems indicated that do not help the reader to understand the specific objective. The relationship between well-being and social network analysis is speculative. It is necessary to be more specific, considering that behavior is one of the domains of animal welfare. Therefore, its study does not guarantee that the other domains can be directly affected.
Regarding the comment regarding the relationship between wellbeing and SNA being speculative, a fair point is raised. While this study aimed to provide insights into animal social behaviours and networks in respect to individual wellbeing, focusing solely on that domain does limit the conclusions that can be drawn regarding welfare. However, we do not believe the current introduction, nor the discussion, implies that social behaviour should be looked at in isolation when measuring wellbeing and make many references to SNA being just one element of a holistic toolbox when assessing the welfare of individual animals. See: “Nonetheless, SNA has potential within animal welfare assessments that consider all aspects of an animal’s life including their social role [3], and in assessing the potential impact of moving animals in or out of an existing social group ahead of time [10].”
The argument is not that SNA predicts wellbeing in and of itself, but rather that it is a helpful tool when looking at the social environment while assessing wellbeing. It is unclear to us where the manuscript suggests otherwise, and so difficult to make amendments which address this comment more specifically than we have done below. It is our belief that there is no one measurement that can definitively tell us whether an animal is experiencing good welfare (which is already a tricky and still rather subjective topic) but this does not preclude examining individual elements of the animal’s lived experience when and where they are important. As the social role is known to be an important element of an animal’s effective state, thus its consideration and inclusion in many modern frameworks of animal welfare including the Five Domains, social relationships and interactions are arguably an important factor influencing psychological and physical health. This is supported by numerous studies, several of which are already referenced within the manuscript.
The following paper also provides a review that is more specific to wild animals: https://www.frontiersin.org/articles/10.3389/fvets.2019.00062/full which is now referenced [4] in the manuscript.
The introduction has been amended to make reference to this paper and address this comment more specifically and hopefully make the intention of the paper in this regard clearer: “As, combined with all other areas of their lives, the social environment is important to physical and psychological health [4] it stands to reason that attention should be paid to measuring the social experience of animals in conjunction with all other areas of their lives when assessing wellbeing.”
A further and more specific reference has also been made in the introduction to hopefully improve the specificity regarding the relationship between social behaviour and wellbeing: “This model emphasises the importance of examining the interface of all areas of an animal’s life, including their interactions with other animals, in influencing mental states [1].”
The following amendments have been made to the discussion, to hopefully make more explicit the intention for SNA to be used in conjunction with other tools that look at other areas of the animal’s lived experience: “Holistic animal wellbeing assessments should incorporate measurements of social wellbeing alongside all other aspects of their physical and psychological health, with an understanding that the social environment can have profound impacts on the mental state of animals alongside all other elements of their lives [1]. Towards this goal, SNA can reveal intricate and subtle relationships between dyads and within groups that traditional measurements of social behaviour rates would not otherwise identify. SNA has valuable potential as part of evidence-based assessments of wellbeing alongside other tools which assess other areas of the animal’s lived experience, following frameworks such as the Five Domains [1].”
The introduction seems like a draft of a review of the topic, rather than a specific context to understand why this study or case study deserves to be published.
Thank you for your comments. The insight gleaned from this comment is that perhaps the manuscript required clearer signposting to indicate the intention of the paper itself, as we believed the reviewer might have misunderstood the purpose of the case study in conjunction with our review of SNA as a tool. The use of the term ‘case study’ was deliberate to this effect as we intend for the discussion to be generalizable across taxa, hence less introduction and discussion on the species specifically as opposed to focusing on the principles underpinning the use of SNA.
The primary purpose of the article, as it is written, was not to describe or make inferences based on the case study itself, though a discussion of it is made to exemplify the points being made. Rather, the purpose of the paper is to first and foremost present how SNA can be used to the benefit of animal welfare assessments, with the case study as an illustration and template for its use in zoos. As such, we do not agree that the current introduction is incomplete or unfocused in respect to this purpose. The introduction has been amended to hopefully make this point clearer: “In the present article, a case study is presented alongside a discussion of the applications of SNA to provide an example of how SNA can be used to benefit the management of even small social groups through an understanding of the complexities of the social dynamics involved in individual groups.”
The simple summary and abstract have also been amended to make the focus clearer.
On the other hand, it highlights some difficulties of research in zoos (lines 112-113); and this is not true base on the large number of papers.
On behalf of the coauthors, I would, respectfully, disagree on this point. As a regular attendant of zoo-based research conferences and working with many zoos and aquariums directly, there is a plethora of individual case-study-based research which never finds its way into peer-reviewed literature – and more research still that is never done at all - owing to primarily methodological concerns (small sample size, too descriptive, unoriginal, etc…) and a lack of recognition of zoo science as a robust science in itself. Many papers that focus on practical animal care and welfare – as opposed to studies of physical health, which tend to have an easier time – indeed go unpublished due to these issues.
While there are indeed zoo studies of this nature which do find their way into the peer-reviewed literature, this is a small fraction of the work done despite the great potential of case studies for the improvement of individual, group, and species-level animal welfare. This discussion can and has formed entire papers. The topic of the challenges of small-scale zoo research has been explored in great detail by other authors, including Geoff and Hosey whose work was referenced within the manuscript: https://www.tandfonline.com/doi/full/10.1080/10888705.2019.1678038?needAccess=true
These lines have been left as-is; we believe further discussion on this topic would is tangential to the manuscript and outside of the scope of the article. We do not consider that the original statements are making unfounded claims in respect of the existing evidence.
Furthermore, I concern the assertion that certain methodologies for the study of behavior are easy to apply by non-entry-level people. Although it may be true that obtaining data does not require much training, the question that motivates its sampling, analysis and discussion is only possible with high training (lines 91-96).
Thank you for raising this comment. It is indeed a fair point to raise that training is required for the in-depth analysis and discussion required for SNA to be useful. Amendments have been made throughout the manuscript, including in the discussion, to highlight more clearly that training is required in supplement for adopting SNA as part of animal welfare assessments, though this is with respect to the fact that none of those roles highlighted in the indicated section (students, visiting academics, caregivers – noting that many zookeepers nowadays have at least Bachelor’s degrees in a biological science) could be considered to be below entry-level for studying the behaviour of animals.
The following section has been added to the discussion to address this point more thoroughly: “While a straightforward method to complete, time, personnel, and expertise are nonetheless required to collect and collate the data, as is relevant training and guidance regarding planning and conducting data collection as well as analysis of the data itself. While the data may be straightforward to collect, a thorough understanding of the social behaviour of the species as well as of analysing data itself is required for any assessment tool to be meaningful.”
We hope this addition addresses your concerns.
The introduction lacks specificity even in the specific objective, negatively affecting the interpretation of the aim of this research. This is another example of the lack of specificity. Furthermore, social stressors are mentioned when they were not previously clearly identified, nor associated with real problems of the study species.
This comment largely connects to the discussion surrounding possible misinterpretation of the intention of the manuscript by the reviewer. When taken as focusing primarily on the applications of SNA rather than a descriptive study of black lemur social behaviour, it is not entirely clear how the objectives of the paper are not specific to the goal of exploring SNA as a tool with generalisable applications.
That said, it is a fair point that specific real problems that SNA can be used to address are not adequately identified in the introduction prior to their inclusion in the discussion. Amendments have been made to hopefully make this clearer earlier in the paper.
Thank you for this comment. As the primary focus of the paper is focusing on the application of social network analysis specifically – and not a descriptive study of black lemur behaviour – shifting the discussion in this direction would fundamentally alter the scope and intention of the manuscript. We disagree with the need to radically convert the manuscript to fulfil a different objective to that which is currently discussed.
SNA, as it is described, is an analysis method based on observations. As such, when discussing animal behaviour it is an observational method in itself. Perhaps this was not made clear enough in the introduction, and as such we have made some alterations which hopefully make this point much more evident early on:
Furthermore, considering the recording rule (lines 185-188) and, for example, the first results presented (lines 219-225), I presume that the authors considered feeding as a possible stressor of the group's social relations. This is not developed, neither in the introduction nor justified in M&M.
Thank you for raising this point. The effect of feeding on social behaviour is now introduced much earlier as part of the introduction as you rightfully point out the first raising of this point was in the discussion. Earlier acknowledgement of this point would have made the later presentation of results and discussion much clearer.
The intention was to ensure that social behaviour was not only looked at in isolation with the understanding that there would likely be differences between feeding and non-feeding times, and that feeding may have been a trigger. We have added the following comment in materials and methods: “As feeding had been associated with increased aggression in previous studies of animal behaviour [11-14], the multiple time frames were selected to ensure social behaviour was not studied in isolation with respect to the fact that aggression may be triggered by husbandry events and not consistent throughout the day.”
We hope these amendments have strengthened the transparency of the article and have made the manuscript more cohesive with a more logical flow of discussion in this regard.
Finally, considering that social structure is founded on behavioral interactions among individuals, and this case study was focused on a family group, a better introduction is necessary to prepare the reader for understanding possible results and differences among studied individuals.
Assuming that the better introduction requested is one that goes into more detail on black lemur social behaviour specifically, our approach to addressing this concern has been twofold. From our perspective, retaining the research focus of presenting SNA as a viable and practical tool was essential. As previously discussed, it is not the intention of this paper to make inferences about nor describe black lemur behaviour but rather to showcase the use of SNA to demonstrate and inspire it’s usage in animal welfare assessment. Muddying this discussion with too much and too detailed a review of black lemur social behaviour would have (1) overcomplicated and confused the existing introduction and (2) significantly bloated the wordcount of the introduction.
This study did not employ similar observational units, there were adults and juveniles.
Thank you for raising this concern. As discussed in the paper, when conducting research in zoos, small sample sizes are a common concern. Very often it is not at all possible to control the demographics of the study population, especially when conducting small-scale case studies such as this one. For such individual cases, it is not possible to simply select a ‘better’ group based on what will have the best statistical power, and certainly, it is not the case in real-world scenarios which is where we are suggesting SNA may be applied.
We do not believe this should preclude zoo research, particularly individual case studies, from being examined. This is particularly true in the context of our manuscript, which does not aim to describe black lemur social behaviour in the context of trying to make inferences about the whole species, but instead provides an illustrative example of how SNA can be used in a supportive and proactive capacity when managing individual animals in human care more broadly.
Additionally, reading the description of the observations I cannot understand the recording methods. It appears that Authors applied a behavior sampling rule and a continuous recording rule. It would not be an instantaneous sampling. This should be reviewed and associated with the core question and/or the experimental design (why did Authors decide to sample before and after feeding?).
Thank you for highlighting this error. Incorrect terminology was used in this instance; it was indeed a continuous sampling method that was utilised for the case study, and the manuscript has been amended to correct this mistake.
I feel that a better description is needed adding information about: attributes of individuals, not only age but also role, dominance rank; types of relationships studied: bonds, dependence, dominance.
It is not clear what the reviewer is referring to here. Under the assumption this comment refers to Table 1 and the associated text in M&M 2.1, we disagree that substantial additional information of this nature is required on the study subjects. The SNA itself reveals and provides evidence for the dominance rank and social role and is described in the results and discussion. Otherwise, any information on bonds and dominance provided in the materials and methods is purely subjective and speculative prior to the data presentation later in the manuscript.
However, with respect to the two juveniles involved in the study, we agree that it is certainly worthwhile to include some information regarding their dependence as this is indeed relevant to the social behaviours observed. The following information has been added: “The juvenile male was no longer dependent on his mother at the time of the study, however, while she was weaned the juvenile female was still partially dependent.”
I consider that the Author´s approach is original for this species and zoo research, however, the lack of focus in the manuscript, the lack of clarity in the sampling rules and of justifications for the structure or sampling periods, do not allow me to accept this proposal to be published.
We appreciate your feedback. We have worked hard on this manuscript and extensively edited it according to the 5 reviewer’s feedback to make it clearer, and to help the readers understand the justification for the points raised. We kindly ask you to reconsider your approach based on the changes made.